# Development of core outcome sets for clinical trials in temporomandibular disorders: A study protocol

**Natália dos Reis Ferreira**[1,2], **Carlos Miguel Machado Marto**[3,4,5,6], **Aleli Tôrres de Oliveira**[2], **Maria João Rodrigues**[1], **Marcos Fabio DosSantos**[2,7,8,9]*

1 Faculty of Medicine, Institute for Occlusion and Orofacial Pain, University of Coimbra, Coimbra, Portugal, 2 Postgraduate Program in Medicine (Radiology), Faculty of Medicine, Federal University of Rio de Janeiro (UFRJ), Rio de Janeiro, Brazil, 3 Faculty of Medicine, Institute of Experimental Pathology, University of Coimbra, Coimbra, Portugal, 4 Coimbra Institute for Clinical and Biomedical Research (iCBR), Area of Environment Genetics and Oncobiology (CIMAGO), University of Coimbra, Coimbra, Portugal, 5 Center for Innovative Biomedicine and Biotechnology (CIBB), University of Coimbra, Coimbra, Portugal, 6 Clinical Academic Center of Coimbra (CACC), Coimbra, Portugal, 7 Laboratory of Mechanical Properties and Cell Biology (PropBio), Prosthodontics and Materials Sciences Department, Federal University of Rio de Janeiro (UFRJ), Rio de Janeiro, Brazil, 8 Postgraduate Program in Dentistry (PPGO), School of Dentistry, Federal University of Rio de Janeiro (UFRJ), Rio de Janeiro, Brazil, 9 Postgraduate Program in Translational Neuroscience (PGNET), Federal University of Rio de Janeiro (UFRJ), Rio de Janeiro, Brazil

* santosmfh@gmail.com, marcos.fabio@odonto.ufrj.br

**Funding:** MFD received fellowships from the following brazilian funding agencies: Conselho Nacional de Desenvolvimento Científico e

## Abstract

### Background

Temporomandibular Disorder (TMD) is a generic term applied to describe musculoskeletal disorders that affect the temporomandibular joint (TMJ), the masticatory muscles and the related structures. TMD comprises two groups of disorders, namely intra-articular TMD and masticatory muscle disorders. There is still difficulty in establishing the effectiveness of different therapeutic modalities for TMD with robust evidence, despite the large volume of publications in the area. The lack of outcomes standardization may represent a limiting factor in the search for scientific evidence.

### Objective

This study aims to develop a core outcome sets (COS) for clinical trials in intra-articular TMD and masticatory muscle disorders.

### Methods

The protocol for determining the COS-TMD will consist of three phases: 1. Synthesis of TMD Management Intervention Outcomes. The identification of outcomes will be carried out through a systematic review, which will include randomized clinical trials that evaluated the effectiveness of interventions used in TMD management. 2. Through a two-round international Delphi survey, the list of outcomes will be scored by three panels of stakeholders. 3. A representative sample of key stakeholders will be invited to participate in a face-to-face meeting where they can discuss the results of the Delphi survey and determine the final core set.

Tecnológico (CNPq) - PQ2 (09/2020) and Fundação de Amparo à Pesquisa do Rio de Janeiro (FAPERJ) - JCNE 2018. Unfortunately, it is not possible unable to secure a letter from the funder confirming my funding award. However, all the related documents have been uploaded as an "Other" file. The funders had and will not have a role in study design, data collection and analysis, decision to publish, or preparation of the manuscript.

**Competing interests:** The authors have declared that no competing interests exist.

## Conclusions

The implementation of this protocol will determine the COS-TMD, which will be made available for use in all TMD clinical studies. The use of COS when planning and reporting TMD clinical trials will reduce the risk of publication bias and enable proper comparison of results found by different studies.

## Introduction

Temporomandibular disorder (TMD) refers to a group of musculoskeletal conditions that affect the temporomandibular joint (TMJ), the masticatory muscles, as well as the associated structures [1]. TMD comprises two main groups of disorders, namely intra-articular TMD and masticatory muscle disorders. Noteworthy, each of those groups comprises different diagnoses of intra-articular TMD and masticatory muscle disorders. The most common TMD diagnoses include arthralgia, myalgia, local myalgia, myofascial pain, myofascial pain with referral, disc displacement disorders, degenerative joint disease, subluxation, and headache attributed to TMD [2]. Furthermore, some of those disorders often coexist in the same subject. Such features contribute to the complex diagnosis and treatment of TMD [2, 3].

TMD affects 5 to 12% of the population. It is considered the second most prevalent group of musculoskeletal conditions, only behind chronic low back pain [4]. Pain, TMJ noises and limited jaw movements are the main clinical findings of TMD [5]. The TMJ noises and limited jaw movements are often associated with intra-articular TMD [6]. On the other hand, pain is a common symptom of different TMD types. It is described as the main complaint that leads individuals to seek treatment. In addition, pain is associated with a greater disability degree [2, 7]. It has been estimated that 30 to 40% of the acute painful TMD cases become chronic [8]. Chronic pain is considered one of the leading health problems. It affects the professional and social activities, the emotional status and the quality of life [5, 7, 9]. Noteworthy, it has been suggested that several factors may participate in the mechanisms of chronic pain of articular origin. For instance, a possible relationship between the osteoarthritis-related pain and the intestinal microbiota has been studied [10]. However, more studies will be necessary to obtain more robust scientific evidence.

There is a wide range of therapeutic modalities used in the management of TMD. They range from conservative therapies such as behavior therapies to surgical treatments [11, 12]. Despite the many therapies applied to intra-articular TMD and masticatory muscle disorder, most of these treatment's effectiveness has not yet been established through robust scientific evidence. Several systematic reviews that have investigated the effectiveness of TMD treatments have concluded that rigorous and high-quality clinical studies are still needed to determine these therapies effectiveness [11–16].

Evidence-based clinical practice is critical to provide quality health care with better results and less risk for patients. To achieve these goals, randomized controlled clinical trials are considered the gold-standard studies since they allow researchers to assess both the safety and the efficacy of different treatments [17]. The combined analysis of randomized controlled trials through systematic reviews and meta-analyses provides more robust scientific evidence. Such approach promotes a significant increase in the analyzed sample size as well as in the quality assessment. Each therapy's efficacy is evaluated through primary and secondary outcomes [18] which provide essential information for decision-making in health care services delivery [19].

Two initiatives have been developed to promote outcomes standardization and to provide an instrument for outcome measurement. Those initiatives are: 1- Core Outcome Measures in Effectiveness Trials (COMET) and 2- Consensus-based Standards for the selection of health Measurement Instruments (COSMIN). The aims of COMET are to promote the development and application of core outcome sets (COS). COS comprises a group of outcomes that must be measured and reported, at least, in all clinical trials of a specific area [20]. On the other hand, COSMIN's objective is to improve the selection of outcome measurement instruments of health research outcomes. This is achieved by developing and encouraging the use of transparent methodology and practical tools to select the most suitable outcome measurement instrument in both research and clinical practice. The main reasons to elaborate COS are to enhance the ability to compare similar studies, reduce publication bias, and increase the relevance of the results obtained in clinical trials and systematic reviews [21]. Therefore, it is essential to determine a core outcome set for the two main TMD groups (e.g., intra-articular TMD and masticatory muscle disorder).

This study protocol aims to evaluate the outcomes of clinical trials that studied the efficacy of conservative or minimally invasive interventions to treat intra-articular TMD and masticatory muscle disorder. The study's purpose is to promote greater clarity and homogeneity in the results presented in these clinical trials. The current protocol use will reduce the publication bias risk and enable a proper comparison of results found by different studies. The use of COS when planning and reporting clinical trials will simplify the determination of the effectiveness of different modalities of therapies used to treat TMD, with high-quality scientific evidence. This strategy will ultimately allow adequate therapeutic decision-making and allocation of health and research resources.

## Materials and methods

This protocol has been developed following the guidelines described in the COMET Handbook [19] and the COS-STAndards for Reporting (COS-STAR) [22]. This study has been registered on the COMET database (https://www.comet-initiative.org/Studies/Details/1830). The COS-STAP Statement (Core Outcome Set-STAndardised Protocol Items) was filled out and is listed in the S1 Table. PRISMA-P (Preferred Reporting Items for Systematic review and Meta-Analysis Protocols) was also filled out and is listed in the S2 Table. A flow-chart illustrating the COS-TMD protocol is presented (Fig 1).

### Ethics statement

This project was evaluated and approved by the ethics committee of the Faculty of Medicine of the University of Coimbra (FMUC) (approval number: 071/2021). All patients involved you will be asked to give explicit (written) consent before participation in the Delphi survey. All procedures will be performed according to the principles of the Declaration of Helsinki.

### Scope determination

COS will be applied in clinical trials that evaluated interventions used to treat TMD. An extensive literature search was performed, including the COMET database, and no article evaluating the outcomes in TMD treatments was found. COS scope will be based on PICO's model (Population; Intervention; Comparison; Outcomes).

**Participants.**

1. Individuals over 18 years old who were diagnosed with TMD of muscular origin (masticatory muscle disorders) according to the Diagnostic Criteria for Temporomandibular Disorders (DC/TMD) or Research Diagnostic Criteria for Temporomandibular Disorders

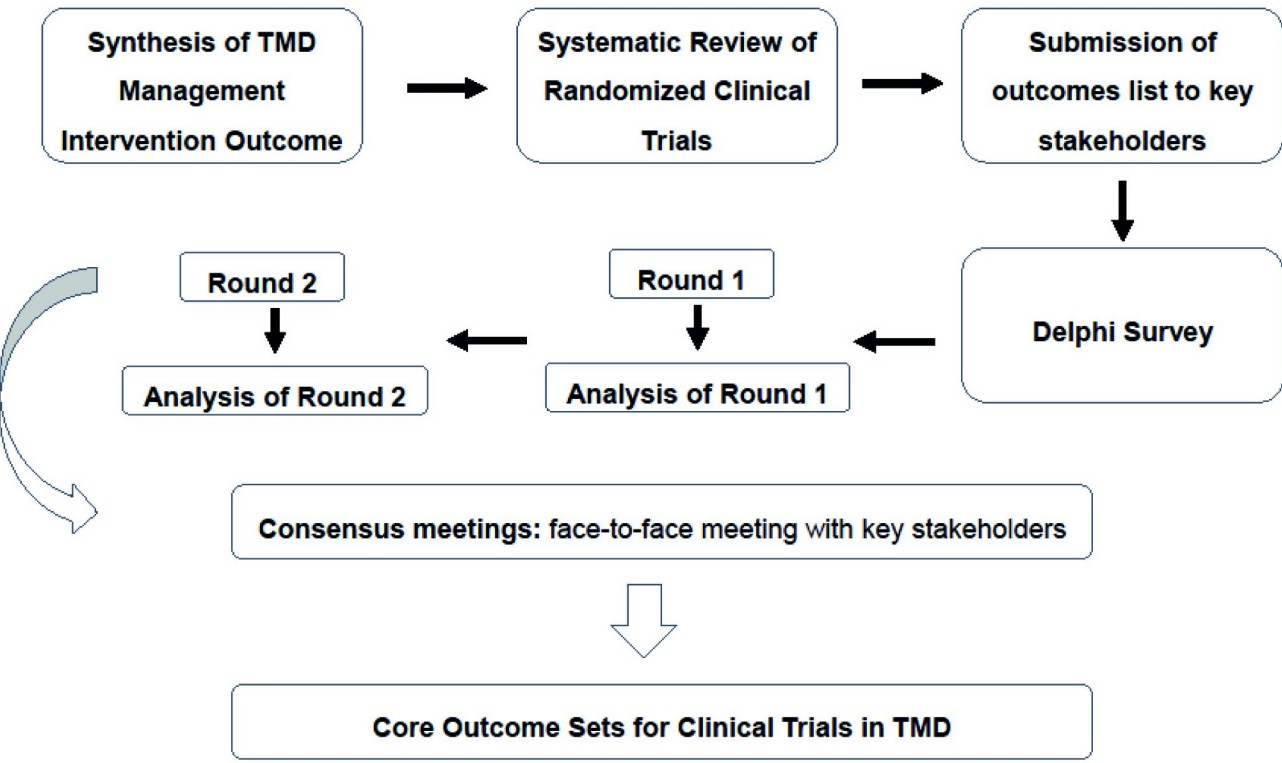

**Fig 1. Flow-chart illustrating the COS-TMD protocol.**

(RDC/TMD), regardless of the disorder severity or the specific diagnoses within the masticatory muscle disorders.

2. Individuals over 18 years old who were diagnosed with intra-articular TMD, according to the DC/TMD or RDC/TMD, regardless of the disorder severity or the specific diagnoses within the group of intra-articular TMD.

**Intervention.**   Any conservative or minimally invasive therapeutic modality used in the management of TMD will be considered.

**Comparison.**   The inclusion of studies that used a placebo or another form of therapy will be considered.

## Synthesis of TMD management intervention outcomes

The outcomes identification will be carried out through a systematic review, which will include randomized clinical trials that evaluated the effectiveness of interventions used in TMD management. The outcomes identified in the systematic review will be categorized in the six domains of the core outcome measures for chronic pain clinical trials, which follows the Initiative on Methods, Measurement, and Pain Assessment in Clinical Trials (IMMPACT) recommendations. The core domains are: (1) pain; (2) physical functioning; (3) emotional functioning; (4) participant ratings of improvement and satisfaction with treatment; (5) symptoms and adverse events; and (6) participant disposition [23, 24]. The systematic review will be registered in the PROSPERO database and will follow the PRISMA statement recommendations.

**Search strategy.**   The search strategy will consist of the combination of terms found in the controlled vocabulary of MeSH (Medical Subject Headings) and synonyms. However, the

search strategy will be adapted to the specifications of each database. The terms will be combined by the Boolean operators "AND" and "OR". Electronic searches will be carried out in the following databases: MEDLINE (via PubMed), Scopus, Web of Science, Cochrane Library, EMBASE and clinicaltrials.org. Electronic searches will be limited to the English language and for the past 5 years. Articles identified in different databases will be imported to the citation manager Endnote, version 20 (Clarivate Analytics, Australia). All references will also be evaluated regarding potential conflicts of interest.

**Article selection.** Two independent researchers will carry out the selection process. In the first phase, all relevant articles will be selected by evaluating only titles and abstracts. The articles selected in the first phase will be read in full. The articles that meet the inclusion criteria will be included in the review. In any disagreements between the two authors take place during the two selection phases, a third author will evaluate the related articles.

The articles will be included using the following inclusion criteria:

**Participants.**

1. Individuals over the age of 18 diagnosed with TMD of muscular origin (masticatory muscle disorders) according to the Diagnostic Criteria for DC/TMD or RDC/TMD, regardless of the disorder severity or the specific diagnoses within the masticatory muscle disorders.

2. Individuals over the age of 18 diagnosed with intra-articular Temporomandibular Disorders according to Diagnostic Criteria for Temporomandibular Disorders (DC/TMD) or RDC/TMD, regardless of the disorder severity or the specific diagnoses within the group of intra-articular TMD.

**Intervention.** Any conservative or minimally invasive therapeutic modality used in the management of TMD will be considered.

**Comparison.** Studies using a placebo, or another form of therapy will be considered.

**Study type.** Randomized clinical trial.

**Obtaining the data.** The following outcomes will be evaluated in this review:

Primary Outcomes

Effectiveness outcomes

Secondary Outcomes:

>   Outcome measurement instruments.

>   Outcome's definitions provided by authors.

Primary and secondary outcomes will be extracted from the selected articles by two independent authors. In addition, data will be collected regarding the number of participants, diagnosis, type of intervention and adverse effects. Some outcomes are likely to be differently defined and measured in the clinical trials included in this review. To overcome this issue, different definitions of outcomes will be put together (extracting the wording description verbatim) under the same outcome name. This will be performed during the first phase of the study.

In a second stage, these outcomes will be categorized according to the domains recommended by the core outcome measures for chronic pain clinical trials: IMMPACT recommendations. Thus, the identified outcomes will be categorized into one of the following domains:

1. Pain.

2. Physical functioning.

3. Emotional functioning.

4. Participant ratings of global improvement.

5. Symptoms and adverse events.

6. Participant disposition.

The two stages used to categorize the outcomes will be carried out by two independent researchers, a methodologist and a specialist in orofacial pain and TMD. A third researcher with extensive experience in pain and methodology will solve the disagreements found.

## Determination of the list of outcomes through the Delphi survey

The list of outcomes from the systematic review will be evaluated by groups of key stakeholders to identify the existence of relevant outcomes that are not included in the initial list. At this stage, three stakeholder groups will be formed. They will be composed of a representative sample of each group involved in the TMD-COS development, namely: health professionals; researchers; and patients.

The compilation of outcomes from the systematic review and stakeholder suggestions could generate a long, ambiguous, and unclear list. Hence, it is necessary to build a clear outcome list for the understanding of participants from different interest groups. It must be concise, balanced and organized to be classified effectively by the participants through the Delphi survey. Also, these groups will be encouraged to assess and suggest changes about clarity, ambiguities, and unnecessary items. After receiving all the suggestions from the key stakeholders, the researchers responsible for developing the COS-TMD will gather to determine the final list of outcomes.

## Delphi survey

The Delphi survey is a tool that consists of an interactive and multi-stage process designed to transform opinion into group consensus. This methodology is employed to enhance decision-making in health and social care [25]. The Delphi survey allows collecting the opinion of several stakeholders online to obtain a consensus. Through the Delphi survey, stakeholders can score the outcomes presented in the questionnaire on an importance scale. The consensus among the various participants is built through feedback from all stakeholders' responses in subsequent rounds. Therefore, the participants can view the opinions of other participants before reevaluating each item and, accordingly, can change their initial responses based on the feedback of previous rounds. Without direct communication between the participants, this feedback provides a mechanism to reconcile the different participants' opinions. An important advantage is that this technique is anonymous, which avoids the effect of dominant individuals. Moreover, it may be distributed to a large number of subjects, regardless of the geographical dispersion. This work will use the Delphi Manager Software.

In this study, there will be three different panels for each group of stakeholders involved in the project. The research team involved in developing the COS-TMD will be composed of three groups of stakeholders: 1- patients with TMD; 2- dentists, doctors and physiotherapists specialized in TMD treatment and 3- researchers. TMD patients will be involved as research partners since the COS need to include outcomes that are most relevant to patients.

The stakeholder who will be part of the patient group will be identified through social media and university clinics. In addition, health professionals will be encouraged to invite their patients. Health professionals specialized in TMD will be identified within national and international professional associations such as the International Association for the Study of Pain, the American Academy of Orofacial Pain, the European Academy of Orofacial Pain and Dysfunction, the European Academy of Craniomandibular Disorders, the Australian/New

Zealand Academy of Orofacial Pain, the Asian Academy of Craniomandibular Disorders and the Ibero Latin American Academy. Researchers will be identified in the relevant publications included in this systematic review and through international research organizations. An effort will also be made to include researchers from different regions of the world. Researchers will be encouraged to invite health professionals from their country to participate in the development of the COS-TMD. The research team is expected to consist of at least 150 stakeholders, with a minimum of 20 participants in each group.

The determination of the COS-TMD will be performed in two rounds to obtain consensus. It is expected that a reduced number of rounds will avoid the dropping out of stakeholders between rounds. This strategy will be important to reduce the risk of attrition bias.

During the questionnaire construction, special attention will be taken to the language used. Lay terms will be preferred over technical terms and medical jargons. As previously mentioned, the questionnaire will be first evaluated by a group of key stakeholders to assess each questionnaire item's clarity. The outcomes will be categorized by the domains proposed by the IMMPACT recommendations. With this respect, the order of the domains in the questionnaire will be pain; physical functioning; emotional functioning; participant ratings of global improvement; symptoms and adverse events; participant disposition. These will be applied in all rounds of the Delphi survey.

**Round 1.**   When participating in the Delphi survey, stakeholders will fill in some demographic data such as age, country of origin and professional activity. On the home page of each Delphi survey round, stakeholders will be provided with a plain-language summary of both COS and Delphi processes. In addition, a plain-language summary of each outcome will be made available. Questionnaires for health professionals and researchers will be presented in English, while the questionnaire for patients will be available in English and Portuguese. A list of all outcomes will be presented before starting the score for each outcome. Then, the following question will be available to all participants: "How important do you think each outcome is to measure the interventions used to manage temporomandibular disorders?"

This work will use a Likert scale as a scoring system to rate the outcomes within the Delphi process. The 9-point Likert scoring system allows researchers to rank the outcomes according to their relevance. Therefore, ratings between 1–3 represent outcomes with limited relevance, between 4 to 6 represent important but not critical outcomes, and scores ranging from 7 to 9 represent critical outcomes. This scoring system is recommended by the Grading of Recommendations Assessment, Development and Evaluation (GRADE) working group to assess the level of importance of research evidence [26].

**Round 2.**   Only participants who have completed round 1 will be able to participate in the second round. In this round, the result of each outcome obtained in round 1, stratified by the stakeholder group, will be presented to the participants. Each participant will be reminded of his classification in round 1. With these data, the participants will be invited to re-evaluate the score of each outcome.

If dropout rates are considered high in the overall assessment or in the assessment provided by a specific group of stakeholders, the following measures will be taken to increase the response rate: personalized reminders and e-mails from distinguished researchers in the area will be sent. Besides, round 2 will be kept open for a more extended period.

**Consensus process.**   A Consensus will be obtained based on the criteria described by Wylde et al., 2015 [27]. The following consensus definition will be used for inclusion in the COS: outcomes that received a score of 7–9 by 70% or more of the participants together with a score of 1–3 by 15% or less of the participants in the three panels, or those with a score of 7–9 by 90% or more in one panel [27]. A Consensus to determine whether an outcome should not be included in the COS will also be based on the 70/15 ratio. However, with 70% or more of

participants scoring the outcome as 1–3 and 15% or less giving a score of 7–9 [28]. Outcomes that do not meet these criteria will be classified as "no consensus achieved".

## Consensus meetings

A representative sample of key stakeholders composed of three groups involved in the Delphi process will be invited to participate in a face-to-face meeting to discuss the Delphi survey results and determine the final core set. Around 20 stakeholders are expected to be gathered at this stage. The results obtained in the second round of the Delphi survey will be sent to the participants for analysis. It will occur one month before the face-to-face meeting.

An experienced facilitator will lead the meeting. In the first phase, each outcome will be discussed, and it will be up to the facilitator to ensure that all participants have the opportunity to express their opinion. After that, an anonymous survey following the same characteristics as the Delphi survey will be conducted. The same criteria applied in the second round of the Delphi survey will be used to define the inclusion or exclusion of the outcome present in the COS. Outcomes that do not meet these criteria will be classified as "no consensus achieved" and will be submitted to further discussion and evaluation.

## Analysis

**1. Outcome scoring and feedback.** The results obtained for each outcome will be analyzed using descriptive statistics and presented to the participants in the next round. The objective is to promote consensus among stakeholders. As this work will use multiple stakeholder panels, the feedback will be calculated separately for each panel. In the subsequent rounds, the participants will receive feedback for each outcome from all panels (identified to which stakeholder group they belong). They will also receive the latest scoring given to the outcome. It is expected that is approach will be important to achieve consensus between groups.

**2. Missing data.** The dropout rate will be assessed in relation to the total number of participants who did not carry out the second round as well as the distribution of these withdrawals among the groups of stakeholders. Also, the potential attrition bias will be assessed through the average scores of individual outcomes among those who have completed or not both rounds. It is expected that this strategy will allow to identify whether there was a pattern or a difference in the score between the participants who completed or not completed the rounds.

## Discussion

The COS development for TMD has not been found in an extensive literature search. Thus, it is essential to determine a core outcome set for the two main groups of TMDs (masticatory muscle disorders and intra-articular Temporomandibular disorders) to promote greater clarity in the results presented in clinical studies, and thus, reduce the risk of publication bias and enable the comparison of the different studies results. The COS adoption when planning and reporting clinical trials will facilitate the determination of the effectiveness of the treatments with high-quality scientific evidence, allowing for adequate therapeutic decision-making and allocation of health and research resources.

## Supporting information

**S1 Table. Core Outcome Set-STAndardised Protocol items: The COS- STAP statement.** (DOCX)

**S2 Table. PRISMA-P (Preferred Reporting Items for Systematic review and Meta-Analysis Protocols) 2015 checklist: Recommended items to address in a systematic review protocol.** (DOC)

## Author Contributions

**Formal analysis:** Natália dos Reis Ferreira, Carlos Miguel Machado Marto, Aleli Tôrres de Oliveira, Maria João Rodrigues, Marcos Fabio DosSantos.

**Writing – original draft:** Natália dos Reis Ferreira, Carlos Miguel Machado Marto, Aleli Tôrres de Oliveira, Maria João Rodrigues, Marcos Fabio DosSantos.

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
