## [Decision Letter · Decision Letter 0]

2 Apr 2022

PONE-D-22-01740Development of Core Outcome Sets for Clinical Trials in Temporomandibular Disorders: A Study ProtocolPLOS ONE

Dear Dr. DosSantos,

Thank you for submitting your manuscript to PLOS ONE. After careful consideration, we feel that it has merit but does not fully meet PLOS ONE’s publication criteria as it currently stands. Therefore, we invite you to submit a revised version of the manuscript that addresses the points raised during the review process.

We look forward to receiving your revised manuscript.

Kind regards,

Marcos Roberto Tovani-Palone

Academic Editor

PLOS ONE

Journal Requirements:

3. Please ensure that you refer to Figure 1 in your text as, if accepted, production will need this reference to link the reader to the figure.

Additional Editor Comments:

This is a relevant and well-written study protocol that aims to develop a core outcome sets (COS) for clinical trials in intra-articular temporomandibular disorder and masticatory muscle disorders. A few suggestions were made by the reviewers and I would like to ask the authors to revise the article.

Reviewers' comments:

Reviewer's Responses to Questions

**Comments to the Author**

1. Does the manuscript provide a valid rationale for the proposed study, with clearly identified and justified research questions?

Reviewer #1: Yes

Reviewer #2: Yes

2. Is the protocol technically sound and planned in a manner that will lead to a meaningful outcome and allow testing the stated hypotheses?

Reviewer #1: Yes

Reviewer #2: Yes

3. Is the methodology feasible and described in sufficient detail to allow the work to be replicable?

Reviewer #1: Yes

Reviewer #2: Yes

4. Have the authors described where all data underlying the findings will be made available when the study is complete?

Reviewer #1: Yes

Reviewer #2: Yes

5. Is the manuscript presented in an intelligible fashion and written in standard English?

Reviewer #1: Yes

Reviewer #2: Yes

6. Review Comments to the Author

You may also provide optional suggestions and comments to authors that they might find helpful in planning their study.

Reviewer #1: I want to congratulate the authors on a very important well designed andmostely needed study in the field of TMD treatment.

Reviewer #2: The authors have developed an interesting and useful study protocol on the outcome sets for clinical trials in intra-articular TMD and masticatory muscle disorders

However, I would like to make some observations before recommending your work for publication.

I recommend the authors in the introduction, when discussing the articular causes of TMD, to mention a possible relationship between the systemic origin of OA and the intestinal microbiota, mentioning the following reference: doi:10.3390/nu13030716.

I believe it is not necessary to include an "Objective" section at the end of the introduction.

7. PLOS authors have the option to publish the peer review history of their article (what does this mean?). If published, this will include your full peer review and any attached files.

Reviewer #1: **Yes: **Emodi-Perlman Alona

Reviewer #2: No

---

## [Author Response · Author response to Decision Letter 0]

4 Apr 2022

Rio de Janeiro, April 04th 2022

Dear Editor, 

We thank very much the reviewers for the helpful suggestions regarding the manuscript titled “Development of Core Outcome Sets for Clinical Trials in Temporomandibular Disorders: A Study Protocol” submitted to Plos One. 

All requested changes have been made and highlighted in the updated version of the manuscript. Please find below specific answers addressing comments from the reviewers.

Journal Requirements 

and

3. Please ensure that you refer to Figure 1 in your text as, if accepted, production will need this reference to link the reader to the figure.

Response: The manuscript has been fully revised in order to match all the journal’s requirements.

Additional Editor Comments:

This is a relevant and well-written study protocol that aims to develop a core outcome sets (COS) for clinical trials in intra-articular temporomandibular disorder and masticatory muscle disorders. A few suggestions were made by the reviewers and I would like to ask the authors to revise the article.

We highly appreciate the editor’s comments. The article has been fully revised. All comments and suggestions have been considered and all comments have been addressed. 

Reviewer’s Comments to the Author 

Reviewer #1: I want to congratulate the authors on a very important well designed and mostely needed study in the field of TMD treatment.

We highly appreciate the reviewer’s comments. 

Reviewer #2: The authors have developed an interesting and useful study protocol on the outcome sets for clinical trials in intra-articular TMD and masticatory muscle disorders However, I would like to make some observations before recommending your work for publication.

We highly appreciate the reviewer’s comments. 

Commmnet #1:

I recommend the authors in the introduction, when discussing the articular causes of TMD, to mention a possible relationship between the systemic origin of OA and the intestinal microbiota, mentioning the following reference: doi:10.3390/nu13030716.

Response: We appreciate the reviewer’s comment. The suggested reference has been added to the manuscript:

11.Sanchez Romero EA, Melendez Oliva E, Alonso Perez JL, Martin Perez S, Turroni S, Marchese L, et al. Relationship between the Gut Microbiome and Osteoarthritis Pain: Review of the Literature. Nutrients. 2021;13(3). Epub 20210224. doi: 10.3390/nu13030716. PubMed PMID: 33668236; PubMed Central PMCID: PMCPMC7996179.

Commmnet #2:

I believe it is not necessary to include an "Objective" section at the end of the introduction.

Response: We have removed this section, following the revoewer’s comment.

---

## [Editor Report · Decision Letter 1]

14 Apr 2022

Development of Core Outcome Sets for Clinical Trials in Temporomandibular Disorders: A Study Protocol

PONE-D-22-01740R1

Dear Dr. DosSantos,

We’re pleased to inform you that your manuscript has been judged scientifically suitable for publication and will be formally accepted for publication once it meets all outstanding technical requirements.

Kind regards,

Marcos Roberto Tovani-Palone

Academic Editor

PLOS ONE

Additional Editor Comments (optional):

The article has been duly revised based on the reviewers' suggestions and is ready for publication.

---

## [Editor Report · Acceptance letter]

19 Apr 2022

PONE-D-22-01740R1 

Development of Core Outcome Sets for Clinical Trials in Temporomandibular Disorders: A Study Protocol 

Dear Dr. DosSantos:

I'm pleased to inform you that your manuscript has been deemed suitable for publication in PLOS ONE. Congratulations! Your manuscript is now with our production department. 

Kind regards, 

on behalf of

Dr. Marcos Roberto Tovani-Palone 

Academic Editor

PLOS ONE